# Role of NF-κB during *Mycobacterium tuberculosis* Infection

**DOI:** 10.3390/ijms24021772

**Published:** 2023-01-16

**Authors:** Nicole Poladian, Davit Orujyan, William Narinyan, Armani K. Oganyan, Inesa Navasardyan, Prathosh Velpuri, Abraham Chorbajian, Vishwanath Venketaraman

**Affiliations:** 1College of Osteopathic Medicine of the Pacific, Western University of Health Sciences, Pomona, CA 91766, USA; 2College of Osteopathic Medicine, Des Moines University, 3200 Grand Ave, Des Moines, IA 50312, USA

**Keywords:** *Mycobacterium tuberculosis*, NF-κB, inflammation, cytokines, granuloma, rifampicin, isoniazid, pyrazinamide, ethambutol

## Abstract

*Mycobacterium tuberculosis* (*M. tb*) causes tuberculosis infection in humans worldwide, especially among immunocompromised populations and areas of the world with insufficient funding for tuberculosis treatment. Specifically, *M. tb* is predominantly exhibited as a latent infection, which poses a greater risk of reactivation for infected individuals. It has been previously shown that *M. tb* infection requires pro-inflammatory and anti-inflammatory mediators to manage its associated granuloma formation via tumor necrosis factor-α (TNF-α), interleukin-12 (IL-12), interferon-γ (IFN-γ), and caseum formation via IL-10, respectively. Nuclear factor κ-light-chain-enhancer of activated B cells (NF-κB) has been found to play a unique mediator role in providing a pro-inflammatory response to chronic inflammatory disease processes by promoting the activation of macrophages and the release of various cytokines such as IL-1, IL-6, IL-12, and TNF-α. NF-κB’s role is especially interesting in its mechanism of assisting the immune system’s defense against *M. tb*, wherein NF-κB induces IL-2 receptors (IL-2R) to decrease the immune response, but has also been shown to crucially assist in keeping a granuloma and bacterial load contained. In order to understand NF-κB’s role in reducing *M. tb* infection, within this literature review we will discuss the dynamic interaction between *M. tb* and NF-κB, with a focus on the intracellular signaling pathways and the possible side effects of NF-κB inactivation on *M. tb* infection. Through a thorough review of these interactions, this review aims to highlight the role of NF-κB in *M. tb* infection for the purpose of better understanding the complex immune response to *M. tb* infection and to uncover further potential therapeutic methods.

## 1. Introduction

### 1.1. Epidemiology of M. tb 

*Mycobacterium tuberculosis* (*M. tb*) infection has exhibited prevalence amongst adults, low- and middle-income countries, and individuals with predisposing health conditions, such as HIV and other immunocompromising diseases [1]. According to reports by the CDC, a total of 7860 cases of tuberculosis (TB) were reported across the United States in 2021, with 526 of these cases being fatal [2]. Despite the U.S. having the lowest rates of TB, 56 cases of multidrug resistant TB were reported in 2020, a public health crisis and threat to the control of TB infection [2]. Globally, on the other hand, 1.6 million individuals reportedly died from TB infection in 2021, designating it as one of the leading fatal infectious agents [1]. Reports by the World Health Organization (WHO) demonstrated the highest number of TB cases in South-East Asia, the Western-Pacific, and Africa, amongst others. Despite low- and middle-income countries accounting for over 80% of reported TB cases, funding for TB diagnosis, treatment, and prevention are insufficient [1]. 

Less than 10% of individuals exposed to *M. tb* exhibit active infection, or primary TB, within two years. Latent infection, on the other hand, is more common amongst exposed individuals, developing without manifesting the clinical signs or symptoms of the disease. Although those with latent *M. tb* are not in danger currently, one study conducted in 2016 utilizing mathematical methods estimated that 1.7 billion individuals developed latent *M. tb* infection, posing a significant risk of potential TB reactivation in such individuals in years to come [3]. 

### 1.2. Inflammation and M. tb Infection 

Shortly after *M. tb* infection, the body elicits a protective immune response against the *M. tb* invader. The primary response of alveolar macrophages and dendritic cells plays a critical role in the subsequent activation of the T-cell response. Granuloma formation, representative of *M. tb* infection, is initiated by the recruitment of lymphocytes and macrophages to the infection site. Interestingly, granuloma formation is dependent on pro-inflammatory mediators of the immune response, such as tumor necrosis factor alpha (TNF-α) [4]. The formation of such granulomas acts to contain the infection and prevent its spread to target tissues. Although inflammation has been shown to potentiate the body’s response to *M. tb* infection, one study demonstrated the positive protective effect of acute inflammation on *M. tb* infection in mice [5]. *M. tb*, however, has developed various mechanisms to evade the immune response, thus, contributing to its pathogenesis. Such mechanisms include the promotion of an anti-inflammatory response, the inhibition of reactive oxygen and nitrogen intermediates, and subverting the effects of phagocytic macrophages [6]. 

### 1.3. Purpose of Our Literature Review

This article aims to provide a comprehensive review of the literature supporting the role of NF-κB in individuals infected with *Mycobacterium tuberculosis*. Moreover, understanding the direct and/or indirect role of NF-κB in *M. tb* infection may lead to the generation of targeted therapies and further understanding of the role that inflammation plays in *M. tb* prognosis and treatment. 

## 2. *M. tb* Pathology

*M. tb* is an acid-fast gram-positive bacterium responsible for causing tuberculosis, an infection commonly of the lungs, that can spread to include various other organs [7,8]. Infection via *M. tb* can result in either active or latent infection, both of which may progress to miliary tuberculosis involving the lymphohematogenous dissemination of *M. tb* that can result in multiorgan dysfunction [9,10,11]. Transmission occurs via respiratory droplets that enter the lungs and infect antigen-presenting cells including alveolar macrophages and dendritic cells, triggering both innate and adaptive immune responses [6,12]. 

As part of the innate response, macrophages and dendritic cells function to phagocytose the bacteria and express pattern recognition proteins (PRPs) such as toll-like receptors (TLRs) [12,13]. TLRs work by recognizing *M. tb* ligands, such as lipoproteins and glycolipids, and function to activate the release of various pro-inflammatory cytokines and chemokines [6,14]. Piergallini et al. studied *M. tb*-induced inflammation within the lungs by analyzing the protein levels of mice infected with *M. tb* via a lipopolysaccharide (LPS) injection. These findings showed increases in tumor necrosis factor (TNF), interleukin-1beta (IL-1β), IL-6, IL-12p70, and IL-10 [5].

The adaptive immune system comes into play as antigen-presenting cells stimulate I T-cells, resulting in their specialization into either CD4+ T-helper (Th) cells or CD8+ cytotoxic T-cells. Both CD8+ T-cells and Th1 cells have been show to secrete interferon-gamma (IFN-γ) in response to *M. tb* infection [14,15,16]. This release of IFN-γ is stimulated by IL-12 release from macrophages and dendritic cells. In turn, IFN-γ functions in the process of stimulating and recruiting more macrophages to the site of infection, and functions as one of the key regulators of granuloma formation [6,17]. 

Initiated and maintained by pro-inflammatory cytokines, granuloma formation occurs from the recruitment of mononuclear phagocytes and lymphocytes working to section-off the site of infection in an attempt to prevent the spread of *M. tb* [6,18,19]. Subsequently, this process allows for *M. tb* survival within the host and maintaining latent infection [6,18]. Ottenhoff et al. found that defects within the IL-12 and IFN-γ activation pathways resulted in increased susceptibility to and severity of *M. tb* infection, along with impaired granuloma formation [17]. They further associated complete IFN-γ receptor 1 (IFN-γR1) deficiency with fatal outcomes in the presence of poorly pathogenic bacterium [17]. Similarly, Flynn et al. found that within IFN-γ knockout mice, macrophage activation was diminished along with the ability to produce appropriate reactive oxygen species (ROS) [15]. 

The granuloma formation seen in *M. tb* infection is noted to be caseating due to their characteristic necrotic core. As granulomas have a hypoxic core to support pathogen containment; Matta et al. discuss the fact that the hypoxic incubation of *M. tb* supports bactericidal acts such as apoptosis, and results in an increase in ROS attempting to promote infected cell death [20]. ROS work to induce cellular damage to macromolecules, such as the double stranded DNA breaks [21]. Upon initial activation through the innate response, alveolar macrophages are able to produce ROS and nitric oxide (NO) in an effort to kill the invading pathogen [20,22,23]. These species play a role in the antibacterial defense against intracellular pathogens, further highlighting the interplay between the innate and adaptive immune responses targeting *M. tb* [20].

This caseum formation is due in part to the presence of lipid-laden macrophages and IL-10 production [19]. Unlike IL-12, IL-10 is noted to function as an anti-inflammatory cytokine, along with the transforming growth factor β (TGF-β) [6]. IL-10 functions via inhibiting the transcription factor NF-κB, which is involved in the transcription of pro-inflammatory cytokines [24]. In a study analyzing murine schistosomiasis, Hoffmann et al. showed that IL-10 plays a role in regulating granuloma-associated tissue fibrosis [25,26]. This supports the idea that maintaining a dynamic balance between pro-inflammatory and anti-inflammatory mediators of infection is needed for a proper immune response to and control of *M. tb* infection [27].

## 3. Role of NF-κB in the Immune System

Through highly organized and specific functions, inflammatory cells play an essential role in identifying foreign invaders and establishing a response to such invaders. Whether such invaders are bacteria, viruses, or toxic chemicals, immune cells are programmed to react with inflammatory processes as a protective response to infection. Nuclear factor κ-light-chain-enhancer of activated B cells (NF-κB) is regarded by many as being the key mediator of the inflammatory process due to its multifaceted role in the inflammatory response. The NF-κB family of transcription factors consists of five structurally-similar proteins: p65 (RelA), p50, c-Rel, RelB, and p52 [28]. NF-κB plays a central role in the pro-inflammatory response via the release of lysosomes in phagosomes and increasing the production of membrane transport molecules to enhance phagolysosome fusion during the invasion of foreign pathogens [29]. As with most processes in the body, the pro-inflammatory role of NF-κB must be properly regulated to avoid the development of acute and/or chronic inflammatory disease and tissue damage. IκB kinase (IKK) inhibitors and their role in preventing IκBα phosphorylation are one such target in anti-inflammatory therapies [30]. 

NF-κB is classified as a family of inducible transcription factors which play a critical role in the mediation of both innate and adaptive immunity, as well as pro-inflammatory responses. Ultimately, NF-κB incorporates the control of two major signaling pathways, deemed canonical and non-canonical, with their respective mechanisms of action to facilitate its downstream effects [31]. The strict regulation of NF-κB is achieved primarily via IκB (an inhibitor of NF-κB), which is therein further regulated by the IKK complex at a phosphorylation level [28]. 

The canonical pathway manipulates this dynamic by the degradation of the inhibitor IκB, thus activating NF-κB and inducing a pro-inflammatory response [32]. The non-canonical pathway facilitates signaling via NIK (NF-κB-inducing kinase), which inhibits IKK, activating the NF-κB pathway via the inducible ubiquitination of p100 [33]. The activation of NF-κB, via either pathway, is dependent on the mammalian expression of diverse classes of pattern recognition receptors (PRRs), such as TLRs (toll-like receptors) and the complement system. PRRs, specifically TLRs, work to recognize the unique features of various microbes, undergoing hetero-dimerization and inducing NF-κB via the extensively studied MyD88-dependent intracellular cascade [34]

The initiation of the NF-κB pathway will result in the activation of M1 and M2 macrophages, which produce the pro-inflammatory cytokines IL-1, IL-6, IL-12, TNF-α and IL-10, IL-13, respectively. Additionally, M1 macrophages induce Th1 (T helper 1) and Th17 (T helper 17) inflammatory T-cell differentiation, further mediating the ongoing inflammatory response [35]. Recent experimentation has yielded the direct connection between NF-κB activation and the subsequent priming of NLRP3 inflammasomes. NLRP3 and related inflammasomes are known to contain caspases, which once active process procytokines IL-1β and IL-18 into their mature, pro-inflammatory forms [36]. 

Prostaglandins (PG), a key mediator of vascular permeability during the inflammatory response, are converted from arachidonic acid via COX (cyclooxygenase)-1 and COX-2 enzymes [37]. PGE2 is produced by prostaglandin E synthase (PGESs), with a microsomal glutathione (mPGES) being functionally coupled with COX-2 [38]. Neuronal COX-2 gene transcription has been identified as a target for upregulation by NF-κB, with a confirmed causal link between the two via concurrent inhibitions by aspirin [39]. IL-1β, a cytokine of NF-κB-stimulated inflammasome origin, has also been established to induce prostaglandin production in the pulmonary cells [40]. In summary, NF-κB occupies a significant role in the regulation of COX-2/mPGES-mediated PG synthesis and the subsequent inflammatory response. 

## 4. Interplay between NF-κB and *M. tb* Infection

There are several mechanisms by which NF-κB affects intracellular signaling and exerts its effects on the immune response against *M. tb*. Certain mechanisms allow for an enhanced immune defense, while other pathways diminish the immune response over the course of the infection. In order to understand the mechanism that reduces the fight against *M. tb*, we must understand that during *M. tb* infection, CD4+ T-cells secrete IL-2 and IFN-γ [41]. Ideally, the secreted IL-2 would continue to increase lymphocytic proliferation to fight against the *M. tb* infection. During the infectious process, *M. tb* induces monocytes to express IL-2 receptors (IL-2R) allowing IL-2 to attach onto these receptors and deter the cytokine from its role in signaling T-cells to proliferate and augment the lymphocytic response [42]. With this in mind, it is interesting to see that *M. tb* induces NF-κB to expand the production of IL-2R by specifically increasing the synthesis of the IL-2Ra chain of the receptor, as a result indirectly decreasing the immune response towards the *M. tb* infection [43]. While this pathway allows *M. tb* to prevail in its infective process, NF-κB also supports the immune system to clear the infection. 

During *M. tb* infection, one of the main cytokines that is produced is TNF-α, which allows for the formation and maintenance of granulomas. In addition, TNF-α attaches onto TNF receptor type 1 (TNFR1) to induce an intracellular signal to activate IκB kinase (IKK) which phosphorylates IκB proteins, leading to their destruction by proteasome-mediated degradation [44]. In a healthy state, NF-κB are latent by being bound to IκB proteins in the cytoplasm of resting cells; thus, once let free, they accumulate and enter the nucleus to mediate the transcription of target genes [44,45]. Some of these genes include the synthesis of additional TNF-α, chemokines, macrophage-activating molecules, and negative regulators of NFκB containing IκBα and A20 [29,46]. While the extent of the effect of these various molecules against *M. tb* is not quite known exactly, Fallahi-Sichani et al. performed a computational model to theoretically understand if this pathway significantly affects the resolution of *M. tb* infection. They concluded that indeed this pathway does play a vital role in restricting the bacterial growth in a granuloma; however, with the caveat that if this pathway is excessively activated, it can lead to increased pathological inflammation of tissue [47]. As a result, this pathway has created for itself a negative feedback mechanism, in which the IκBα and A20 that are produced decrease the activity of TNFR1, and thus decrease IKK activation, leading to an attenuated NF-κB signaling, as shown in Figure 1 [48]. This feedback mechanism led Fallahi-Sichani et al. to propose that the balance of NF-κB activation leads to the containment of bacteria within a granuloma, and that this balance may be optimally changed to allow an existing T-cell-mediated response to possibly clear the bacterial infection [47]. Some of these effects and mediators surrounding NF-κB have been summarized in Table 1 below. 

In experiments conducted within murine models, Kumar et al. further discovered that A20 is also a target of the let-7 family of microRNAs (miRNA). Let-7f miRNAs, secreted by macrophages, target A20 to modulate NF-κB activity during *M. tb* infection. In the *M. tb*-infected macrophages of mice, the expression of let-7f decreases and the expression of A20 increases, allowing for the progression and survival of *M. tb* [49]. To verify this finding, RAW264.7 macrophages transfected with let-7f mimics prior to *M. tb* infection of the murine models resulted in compromised *M. tb* survival following infection due to the inhibition of A20. Similarly, following *M. tb* infection in RAW264.7 macrophages transfected with a let-7f inhibitor, the A20 levels were augmented, resulting in increased bacterial survival and decreased NF-κB activity [49].

Contrary to the effects of NF-κB activation against *M. tb* infection, Bai et al. found that the inhibition of NF-κB also decreases the survival of *M. tb* in isolated human macrophages, including differentiated THP-1 monocytes, primary monocyte-derived macrophages, and alveolar macrophages. When THP-1 cells were pre-incubated with BAY, a specific IKK inhibitor, and then incubated with *M. tb*, there was a 79% increase in apoptosis following four days of incubation and a 34% increase in apoptosis following eight days of incubation when compared to uninfected cells. These effects resulted in decreased intracellular *M. tb* recovery [50]. In addition, Bai et al. found that the inhibition of NF-κB also enhanced the autophagy of infected THP-1 cells, thereby also reducing the number of viable intracellular *M. tb* recovered from the infected macrophages [50].

Recently, Xia et al. performed an experiment on the effect of *M. tb* on NF-κB and found a mechanism in which the bacteria reduces NF-κB signaling [51]. *M. tb* contains a gene called Rv0927c, which is 792 base-pairs long and may code for a short dehydrogenase/reductase that plays a role in the production of mycotic acid in the cell wall [52]. The experiment by Xia et al. includes in vivo and in vitro analyses of the effect of this gene against NF-κB and found that it reduces its signaling pathway. In addition to reducing NF-κB signaling, they found that *M. tb* growth increased as a result of this effect [51]. Thus, it would be interesting to know if the inhibition of this pathway caused by the Rv0927c gene would allow for greater NF-κB activation and the enhanced clearance of *M. tb* infection.

## 5. Effects of *M. tb* Drugs on NF-κB

### 5.1. Common Drug Therapy Used to Target M. tb

One factor that makes *M. tb* such a pathogenic threat is its developing resistance to prescription medications [50]. Currently, the standard *M. tb* treatment includes a regimen of four drugs: isoniazid (INH), rifampicin (RIF), pyrazinamide (PZA), ethambutol; jointly referred to as RIPE therapy [8,53]. Classically, for cases uncomplicated by drug resistance, a common recommended treatment regimen is the six-month regimen marked by the administration of all four drugs for a period of two months, followed by the continued administration of isoniazid and rifampicin for an additional four months [54,55,56].

### 5.2. Isoniazid on NF-κB

The drug isoniazid is activated by the mycobacterial enzyme catalase-peroxidase (KatG), and functions to inhibit the production of mycolic acid, which is essential for the generation of the mycobacterial cell wall [57]. To study the anti-inflammatory properties of INH, Zhang et al. used a zebrafish model by exposing them to various concentrations of INH for 72 hours and then measuring the resulting migration and accumulation of inflammatory cells [58]. Through the use of a real-time polymerase chain reaction (RT-PCR) they were able to measure the levels of transcription for inflammation-related genes. In doing so, they found that INH administration resulted in a decrease in inflammation and a decrease in the transcription of NF-κB, as shown in Figure 2. This topic of research is not yet fully understood, as contradicting results have also been presented suggesting that INH actually plays a role in the activation of NF-κB [59,60]. Using an electrophoretic mobility assay, He et al. found that the co-administration of INH and RIF resulted in the upregulation of NF-κB DNA-binding activity within the liver. One complication of anti-tb drug therapy is hepatotoxicity and He et al. suggest that this is associated with the oxidative injury caused by INH and RIF, along with the activation of NF-κB [59,60]. They conclude that pyrrolidine dithiocarbamate (PDTC), a potent and known inhibitor of NF-κB, can actually help protect against the hepatotoxicity caused by anti-tb medications such as isoniazid and rifampicin [60,61].

### 5.3. Rifampicin Effect on NF-κB

Rifampicin targets *M. tb* infection through its action of binding the bacterial DNA-dependent RNA polymerase, halting RNA synthesis [62,63]. In order to better understand the effects of RIF on NF-κB, Kim et al. studied the LPS-activated RAW264.7 cell of murine macrophages (KIM). Contrasting with the previously discussed findings of He et al., which suggested that the co-administration of INH and RIF increased NF-κB activation, Kim et al. found that rifampicin inhibits LPS-induced TLR-2 through the action of suppressing NF-κB DNA-binding activity, as NF-κB is a major transcription factor involved in the regulation of TLR-2 [60,64]. Bi et al. came to a similar conclusion in their study of the neuroprotective effects of rifampicin via its ability to inhibit pro-inflammatory mediators. The results of this study suggest that rifampicin supports NF-κB suppression by blocking the degradation of inhibitor IκB proteins, as depicted in Figure 2.

### 5.4. Pyrazinamide Effect on NF-κB

Pyrazinamide, a nicotinamide analog, works by pyrazinoic acid’s disruption of the *M. tb* membrane transport mechanisms. Although not fully understood, the antibiotic effects of this drug are believed to function by targeting the inhibition of multiple processes, such as energy production, trans-translation, and coenzyme A [65,66]. Studying the effects PZA has on the release of various cytokines and chemokines of *M. tb*-infected human monocytes, Manca et al. found PZA to reduce the release of these pro-inflammatory mediators [67]. Through the microanalysis of mouse lungs infected with *M. tb*, they found a reduction in the expression of genes involved within the NF-κB pathways. They hypothesize that the reduction of pro-inflammatory cytokines is associated with the peroxisome-proliferator activated receptor, which has anti-inflammatory effects, and NF-κB-dependent pathways [67]. Studying NF-κB expression after the oral administration of INH, RIF, PYZ, and a combination of all three medications, Yuhong found that, within the RIF, PYZ, and triple therapy, there was initially an increase and then a decrease in NF-κB expression with the prolongation of administration time, as shown in Figure 3, unfortunately without an exact time scale known [68]. From these measurements, the initiation of triple therapy had the largest increase in NF-κB stimulation. Similar to He et al., they conclude that there is increased NF-κB activation in mouse models with liver injury from anti-tb drug administration.

### 5.5. Ethambutol Effect on NF-κB

Ethambutol is known to function by blocking arabinosyl transferase, ultimately inhibiting the carbohydrate formation needed at the level of the cell wall [69]. Although used with RIPE drug therapy to target *M. tb* infection, the effects of ethambutol on NF-κB specifically is an area of research that has yet to be explored [8,53].

## 6. Future Studies

As a result of the opposing regulations of NF-κB during an infection such as *M. tb*, further experiments could be conducted to hone in on the optimal level of NF-κB within the immune system to allow for the ideal fight against *M. tb* infection. Furthermore, since the anti-tb drugs, RIPE, have opposing effects on the level of NF-κB activation, one could look into the possibility of varying these drugs to establish the optimal level of NF-κB found. In this way, it would be possible to treat *M. tb* with anti-tb drugs, as well as enhance the host’s own immune system to fight against the infection. Another possible benefit of this would be if the drug levels used to optimize NF-κB happened to be less than the standard dosages used during active *M. tb* infection, as the host would then also suffer less from the hepatotoxic effects of these drugs. In addition, since there is very little information on the effect of ethambutol on NF-κB, more studies could be conducted to shed light on its effect on NF-κB. In addition to exploring the benefits of NF-κB during *M. tb*, it would be enlightening to study the causes and effects of NF-κB during the infectious processes of a variety of pathogens in order to compare and contrast the capabilities of NF-κB and its effect on the immune system, as very little is confidently known about it.

## 7. Conclusions

NF-κB’s classification as a pro-inflammatory cytokine does not fully evaluate the complete role that it encompasses in *M. tb* infection. Different interactions between the host cells, such as the alveolar macrophages, cells involved with the complement pathway, CD4, CD8, and other antigen presenting cells, dictate whether NF-κB is activated or repressed. The persistence of *M. tb* for generations and its prominence as one of the deadliest and easily communicable diseases highlights the resistance it exhibits against host immune responses. NF-κB has been shown to be modulated by various mechanisms due to its significant role in both adaptive and innate immunity. As the activation of both canonical and noncanonical pathways are dependent on PRRs such as TLRs, a vital area for therapeutic interventions could be centered around the specific activation of TLRs to induce NF-κB’s protective responses against infections, such as its downstream activation of M1 and M2 macrophages, Th1 and Th17 cells, inflammasomes, and pro-inflammatory cytokines that will help aid the formation of granulomas and help to contain and decrease the bacterial load. However, due to the complex regulatory mechanisms set around NF-κB, the system’s ability to actively fight an organism can be allowed to induce a T-cell mediated clearance of the bacteria. While NF-κB has been shown to lower the T-cell response by its effect on the synthesis of IL-2R, NF-κB with the optimal activation has been depicted to lower the bacterial load and contain a granuloma. Due to the increase of infection during both increased and decreased NF-κB signaling, NF-κB has an innate negative feedback mechanism through the transcriptional activation of IκBα and A20, which will indirectly decrease the activation of IKK, leading to attenuated NF-κB signaling. A20 was also discovered to be regulated by *M. tb’s* inhibition of Let-7f miRNAs, which would increase the concentration of A20, further downregulating NF-κB. However, the downregulation of NF-κB did not necessarily prompt an increase in the survival of *M. tb* through the increased apoptosis and autophagy of infected cells. According to current clinical guidelines, a four-medication regimen of isoniazid, rifampin, pyrazinamide, ethambutol is recommended. However, these pharmacological modulators do reflect an effect on the production of NF-κB, with isoniazid, rifampin, and pyrazinamide all producing anti-inflammatory effects by suppressing the production or effect of the NF-κB pathway. Although there is room for further study on the topic of the effects RIPE treatment has on NF-κB, studies suggesting the RIPE suppression of NF-κB activation align with research findings that the inhibition of NF-κB decreases *M. tb* survival. This sheds light on its importance as a key cytokine that can possibly stand as a key therapeutic target for the imposed immune-resistance pathways that *M. tb* uses, once an optimal signaling strength is found. This deep dive into the regulation, relationships, and roles of NF-κB can provide better strategies for ameliorating active and latent *M. tb* infections.

## Figures and Tables

**Figure 1 ijms-24-01772-f001:**
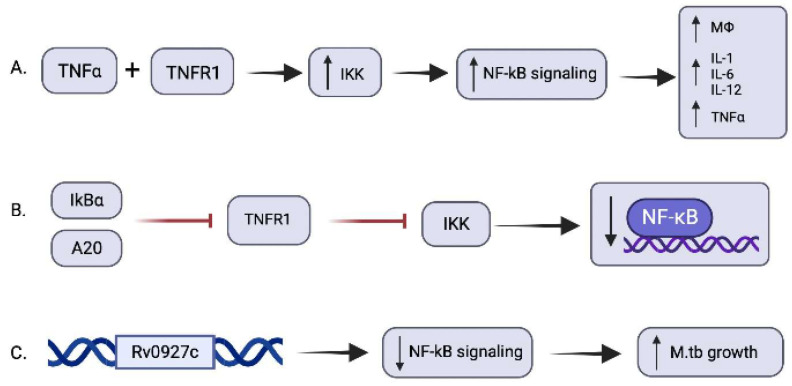
Signaling pathways revolving around NF-κB and the effect of *M. tb*. (**A**) TNF-α binds with TNFR1 to activate IκB kinase leading to increased free NF-κB signaling to produce various pro-inflammatory cytokines such as IL-1, IL-6, IL-12, TNF-α, and activated macrophages. (**B**) The additional products of NF-κB containing IκBα and A20 result in the inhibition of TNFRI, an activator of IKK. The inhibition of TNFR1 and the subsequent inhibition of IKK leads to a decrease in NF-κB. (**C**) The Rv0927c gene activated during an *M. tb* infection reduces NF-κB signaling and, in turn, increases *M. tb* growth and survival as a result.

**Figure 2 ijms-24-01772-f002:**
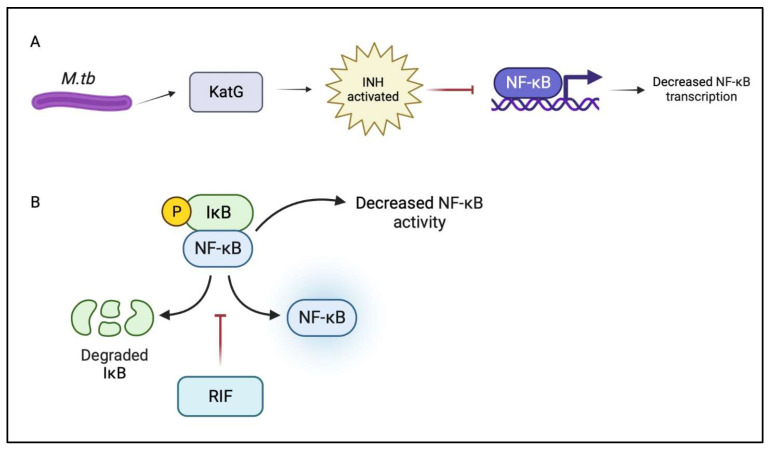
(**A**) During a tuberculosis infection, *M. tb* releases catalase-peroxidase (KatG) in order to assist itself in its own survival; however, in the presence of isoniazid, it activates the drug. One study showed that activated isoniazid in turn decreases the transcription of NF-κB which would reduce its activity and potential as an anti-inflammatory agent. (**B**) One possible effect of rifampicin on NF-κB is the ability of the drug to stop the degradation of IκB proteins, which in a healthy state keep NF-κB inactive. As a result, by reducing the degradation of IκB proteins, less NF-κB is released and thus its activity is reduced.

**Figure 3 ijms-24-01772-f003:**
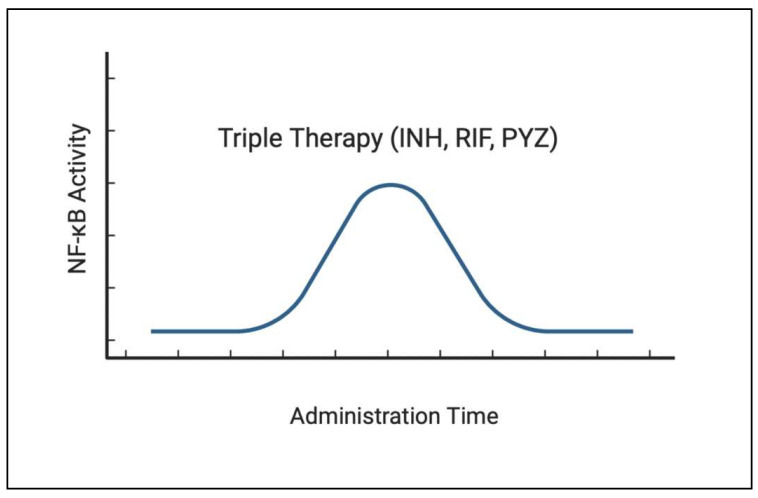
When treating tuberculosis infection with triple therapy, which includes isoniazid, rifampicin, and pyrazinamide, the level of NF-κB initially rises, which could assist with the clearance of the infection. However as the administration time continues with triple therapy, the level of NF-κB subsequently begins to decline after some time.

**Table 1 ijms-24-01772-t001:** A summary of methods that increase and decrease NF-κB activity. As well some of the mediators of NF-κB’s activity. In addition, some of the effects of NF-κB towards *M. tb* infection.

Summary of the Role of NF-κB during *M. tb* Infection	
Cause and effect of increased NF-κB and its enzymes	Cause and effect of decreased NF-κB and its enzymes
- An increase in NF-κB to a certain unknown level has been shown to be effective as an anti-inflammatory mediator to assist in the fight against tuberculosis - NF-κB increases IL-2R production and thus reduces the lymphocytic response towards *M. tb*, thus a substantial increase in NF-κB has been shown to increase the survival of *M. tb* - Triple therapy of INH, RIF, PYZ has been shown to be the most effective combination of anti-tb drugs to increase NF-κB - The activation of TNFR1 activates IκB kinase which phosphorylates IκB proteins to release free NF-κB	- INH, RIF, PYZ administered individually seem to decrease NF-κB activity - Decreased NF-κB has been shown to allow increased *M. tb* survival - NF-κB remains inactive by being bound to IκB proteins - IκBα and A20 reduce TNFR1 activity, which, in turn, reduce IκB kinase and subsequently reduce free NF-κB

## Data Availability

Not applicable.

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
