# Peer review of "Role of NF-κB during Mycobacterium tuberculosis Infection"

_ijms, 2023, doi:10.3390/ijms24021772_

Round 1

Reviewer 1 Report

In the manuscript entitled “Role of NF-?B and Inflammation during Mycobacterium tuberculosis Infection”, the authors reviewed and highlighted the role of NF-κB in Mycobacterium tuberculosis infection.

As a review article for a reputable journal, the manuscript has many imperfections and is relatively short. It must be thoroughly revised and revised for consideration.

1.        The title should be revised as NF-B itself has a pro-inflammatory response to chronic inflammatory disease; thus, the statement "Role of NF-B and Inflammation" appears to refer to two distinct entities. In my opinion, 'Role of NF-B' (omitting 'inflammation') is more appropriate here.

2.        Reference are not formatted according to journal guidelines. Check the abbreviations.

3.        This is a review article, so writing ‘we will investigate’ and ‘Through identification of these interactions’ seem not much appropriate.

4.        Revise the sentence “Concurrently, we will analyze the effects of current therapeutic medical recommendations on NF-κB functioning to target M. tb infection.” Here “we will analyze” should not be used.

5.        In general, a visual or graphical representation is more engaging than the word portion of a review article. To enhance the quality of the article, the authors should add a few additional figures and tables.

Author Response

Dear Reviewer,

We appreciate the time and effort you have all taken to provide an extensive and thorough review of our manuscript. We are also very encouraged by how well it was accepted for the most part and we agree that the suggested edits have greatly improved our manuscript. The objective of our paper was to provide a brief discussion of the effect of NF-κB during M. tb infection and how it may or may not assist the immune system to fight the infection. We hope that our revised manuscript based on all of the suggested edits make this objective much clearer, and we look forward to any further suggestion you may have to bring this manuscript to its full potential.

Thank you all for you valuable time.

 REVIEWER#1

 In the manuscript entitled “Role of NF-?B and Inflammation during Mycobacterium tuberculosis Infection”, the authors reviewed and highlighted the role of NF-κB in Mycobacterium tuberculosis infection.

As a review article for a reputable journal, the manuscript has many imperfections and is relatively short. It must be thoroughly revised and revised for consideration.

  1. The title should be revised as NF-B itself has a pro-inflammatory response to chronic inflammatory disease; thus, the statement "Role of NF-B and Inflammation" appears to refer to two distinct entities. In my opinion, 'Role of NF-B' (omitting 'inflammation') is more appropriate here.

Thank you for this suggestion as we understand and see the issue you have pointed out. We will gladly omit the word inflammation in order to provide a better understanding title.

  1. Reference are not formatted according to journal guidelines. Check the abbreviations.

We attempted to fix the references using endnote as the guidelines state and have compared it to articles submitted to this journal which seem to be identical to our eyes however if it continues to be incorrect, please let us know the type of format you are specifically looking for. Apologies if there is any misunderstanding on this, thank you for your input.

  1. This is a review article, so writing ‘we will investigate’ and ‘Through identification of these interactions’ seem not much appropriate.

Much appreciated for this comment as we see the issue of using these words in a review article as it can be misleading. It has been corrected to “Through a thorough review of these interactions”

  1. Revise the sentence “Concurrently, we will analyze the effects of current therapeutic medical recommendations on NF-κB functioning to target M. tb infection.” Here “we will analyze” should not be used.

Thank you for pointing these misleading phrases as it helps our manuscript be more understanding. We have fixed this error omit the statement starting with “Concurrently” and have changed the second phrase to “we will discuss” as it is more appropriate.

  1. In general, a visual or graphical representation is more engaging than the word portion of a review article. To enhance the quality of the article, the authors should add a few additional figures and tables.

We agree with your input of visual representation is a hallmark of review articles and allows easier understanding, thus we have added 2 more figures and a table summarizing some of the information discussed. Thank you for your suggestions!

Reviewer 2 Report

In this review, the authors have discussed about the different immune response strategies adopted by the host when infected with Mycobacterium tuberculosis. Mycobacterium tuberculosis is a global threat, killing ~ 2 million people worldwide annually. It is estimated that one-third of the human population is carrier of the pathogen. Once the pathogen enters the host, it can result in one of the two-outcomes: it can cause active infection, or it can transform itself to a latent/dormant state. In the latter case, it can spring into the activity when the host is immune compromised either due to age or due to another comorbidity. The transcription factor NF-KB of the host plays a key role in the attenuation of the pathogen by activating macrophages and T-cells (both cytotoxic as well as the helper T-cells). The authors have also discussed about the effects of drugs like Rifampicin, Isoniazid and Pyrazinamide on increasing the level of NF-kB. Overall, this is a major area of research, and a well written review should of interest to a lot of researchers involved in studying Mtb pathogenesis.

I do have couple of suggestion for the authors that I’ve outlined below:

Major Comments

1.     The authors have not discussed any new ideas or strategies Mtb can use to overcome the host-pathogen interaction. The review should always incorporate some novel ideas from the authors on the topic. However, I felt that the authors have compiled all the relevant information on this topic and presented it as a bucket list. The authors should allocate a section on this.

2.     The review needs arrangement. For example, after discussing about NF-kB and its role in tuberculosis, the authors introduce the Mtb pathology. In my opinion the Mtb pathology section should come fast followed by the role of NF-KB in tuberculosis.

3.     Role of Reactive oxygen species in the granuloma can be discussed in details. I felt like this section need little work.

4.     One thing the authors could introduce is the phylogeny of NF-KB across different pathogens as phylogenetic analysis can often give a broader picture of pathogenicity.  

Minor comments:

1.     Line 16: Replace “a” with alpha.

2.     Line 17: nuclear should be capitalized.

3.     Line 62: replace TNF-a with TNF-alpha.

Author Response

Dear Reviewer,

We appreciate the time and effort you have all taken to provide an extensive and thorough review of our manuscript. We are also very encouraged by how well it was accepted for the most part and we agree that the suggested edits have greatly improved our manuscript. The objective of our paper was to provide a brief discussion of the effect of NF-κB during M. tb infection and how it may or may not assist the immune system to fight the infection. We hope that our revised manuscript based on all of the suggested edits make this objective much clearer, and we look forward to any further suggestion you may have to bring this manuscript to its full potential.

Thank you all for you valuable time.

Reviewer #2

In this review, the authors have discussed about the different immune response strategies adopted by the host when infected with Mycobacterium tuberculosis. Mycobacterium tuberculosis is a global threat, killing ~ 2 million people worldwide annually. It is estimated that one-third of the human population is carrier of the pathogen. Once the pathogen enters the host, it can result in one of the two-outcomes: it can cause active infection, or it can transform itself to a latent/dormant state. In the latter case, it can spring into the activity when the host is immune compromised either due to age or due to another comorbidity. The transcription factor NF-KB of the host plays a key role in the attenuation of the pathogen by activating macrophages and T-cells (both cytotoxic as well as the helper T-cells). The authors have also discussed about the effects of drugs like Rifampicin, Isoniazid and Pyrazinamide on increasing the level of NF-kB. Overall, this is a major area of research, and a well written review should of interest to a lot of researchers involved in studying Mtb pathogenesis.

I do have couple of suggestion for the authors that I’ve outlined below:

Major Comments

  1. The authors have not discussed any new ideas or strategies Mtb can use to overcome the host-pathogen interaction. The review should always incorporate some novel ideas from the authors on the topic. However, I felt that the authors have compiled all the relevant information on this topic and presented it as a bucket list. The authors should allocate a section on this.

Thank you for pointing this out, as it is an important section of review articles to shed light into new areas of research. We have allocated a paragraph before the conclusion discussing future studies to enlighten us more about the effect of NF-kB on M. tb

  1. The review needs arrangement. For example, after discussing about NF-kB and its role in tuberculosis, the authors introduce the Mtb pathology. In my opinion the Mtb pathology section should come fast followed by the role of NF-KB in tuberculosis.

Much appreciated for this comment, as we see the issue of the order we originally had. The manuscript has a much better flow of information with your suggested re-arrangement and it has been done so. Thank you for this input!

  1. Role of Reactive oxygen species in the granuloma can be discussed in details. I felt like this section need little work.

Thank you for this suggestion, and thus more has been added to this section and we believe it has strengthened our manuscript and helped to delve into a deeper discussion. Much appreciated!

  1. One thing the authors could introduce is the phylogeny of NF-KB across different pathogens as phylogenetic analysis can often give a broader picture of pathogenicity.  

This suggestion was genuinely very interesting as we can see the importance and effectiveness of adding such information to our manuscript. Unfortunately, with further research into the topic we had much difficulty finding accurate and relevant material surrounding the topic of phylogeny of NF-kB and its effect in a variety of pathogens that could relate to M. tb. As a result of unsuccessfully finding relevant information, we decided to added this to our future studies section as it is without doubt an important topic to discuss when speaking about the effect of NF-kB in any infection. Thus, we believe it made an excellent addition to mention as an area of research that could be conducted. Thank you for bringing this to the forefront, much appreciated!

Minor comments:

  1. Line 16: Replace “a” with alpha.
  2. Line 17: nuclear should be capitalized.
  3. Line 62: replace TNF-a with TNF-alpha.

Thank you for pointing these errors out, as much as we attempt to edit somethings go past our eyes. Much appreciated as this allowed us to find and fix several other typos and errors.

Reviewer 3 Report

Although not many review papers have reported on NF-kB function in Mycobacterium tuberculosis Infection, this paper showed too narrow point only regarding NF-kB.

Therefore, this reviewer strongly suggests that authors should include all enzymes related to the activation or inhibition of NF-kB pathway under Mt infection conditions. 

Signaling pathway with target enzymes of the drugs mentioned in the manuscript should be drawn as figures.

A lot of typo erros (eg., symbolic words) are found in the manuscript. Check all

Summary figure should be included in Conclusion section.

Author Response

Dear Reviewer,

We appreciate the time and effort you have all taken to provide an extensive and thorough review of our manuscript. We are also very encouraged by how well it was accepted for the most part and we agree that the suggested edits have greatly improved our manuscript. The objective of our paper was to provide a brief discussion of the effect of NF-κB during M. tb infection and how it may or may not assist the immune system to fight the infection. We hope that our revised manuscript based on all of the suggested edits make this objective much clearer, and we look forward to any further suggestion you may have to bring this manuscript to its full potential.

Thank you all for you valuable time.

Reviewer #3

Although not many review papers have reported on NF-kB function in Mycobacterium tuberculosis Infection, this paper showed too narrow point only regarding NF-kB.

  • Therefore, this reviewer strongly suggests that authors should include all enzymes related to the activation or inhibition of NF-kB pathway under Mt infection conditions. 

Thank you for this suggestion. As NF-kB is not a well-studied topic there are limitations in the enzymes and relevant factors surrounding it. However, we have attempted to add as many relevant enzymes and mediators that are relevant and have additionally highlighted them in the summary table before the conclusion for readers to be aware of.

  • Signaling pathway with target enzymes of the drugs mentioned in the manuscript should be drawn as figures.

Thank you for this excellent suggestion as pictures speak louder than words. We have added 2 additional figures depicting some of the general pathways that are discussed relating to the NF-kB activity and anti-tb drugs used. Greatly appreciated!

  • A lot of typo erros (eg., symbolic words) are found in the manuscript. Check all

We have reviewed it once more and you are correct that there were several typos and errors in the manuscript. We believe we have corrected them all. Much appreciated for pointing this out in order for us to strengthen the writing of our manuscript.

  • Summary figure should be included in Conclusion section.

Excellent suggestion, especially for a review article as a summary figure will undoubtedly strengthen the information we attempted to depict. A summary table has been added and we hope you find it desirable. Much appreciated for this suggestion!

Round 2

Reviewer 1 Report

The authors successfully addressed the comments raised and made the necessary revisions.

Reviewer 3 Report

Authors have fully addressed all my issues. So, this new version is now acceptable.